# The Synergistic Impact of Crystal Seed and Fluoride Ion in the Synthesis of Silicalite-1 Zeolite in Low-Template Systems

**DOI:** 10.3390/ma17010266

**Published:** 2024-01-04

**Authors:** Xiaojing Meng, Yanjie Qin, Yang Zhang, Min Li, Huibang Huang, Jiaqin Peng, Liangxu Zhou, Jian Feng

**Affiliations:** College of Chemistry and Chemical Engineering, Chongqing University of Science and Technology, Chongqing 401331, China; 2021205157@cqust.edu.cn (Y.Q.); 2022205029@cqust.edu.cn (Y.Z.); 2021205163@cqust.edu.cn (M.L.); 2022205048@cqust.edu.cn (H.H.); 2021205160@cqust.edu.cn (J.P.); 2023205045@cqust.edu.cn (L.Z.); 2021205167@cqust.edu.cn (J.F.)

**Keywords:** Silicalite-1, crystallization mechanism, NH_4_F, S-1 seeds, morphology

## Abstract

Silicalite-1 zeolites are widely applied in gas adsorption, catalysis, and separation due to their excellent hydrothermal stability and unique pore structure. However, traditional preparation methods have inherent drawbacks such as high pollution, high cost, etc. Therefore, this work proposed a green and efficient route for preparing Silicalite-1 zeolite by adding NH_4_F (F/Si = 0.1) and seeds (10 wt%) in a much shorter time (8 h) in a low-template system (TPA^+^/Si = 0.007). It was found that NH_4_F is beneficial for inhibiting the formation of SiO_2_. The S-1 seeds could drastically induce the formation of the zeolite skeleton structure. Noteworthy, the morphology of zeolites was determined by the relative content of NH_4_F and seeds. The crystal morphology is determined by the higher content of the two substances; however, when the content is similar, the crystal morphology is determined by NH_4_F. The results showed that simultaneous control of NH_4_F and seeds can suppress SiO_2_ formation, can improve the relative crystallinity of products, and can be precisely regulated via the synergistic effect of both in zeolite morphology. This work not only provides new ideas for regulating the morphology of silicate-1 crystals but also offers a new path for industrial large-scale production of low-cost and efficient zeolites.

## 1. Introduction

Silicalite-1 zeolite, with its unique pore structure, excellent stability, lipophilicity, hydrophobicity, selective catalytic properties, and other excellent physicochemical properties compared to natural zeolites, is considered as an ideal inorganic material in industry. It has a wide range of applications in the fields of the selective catalysis of aromatic compounds, adsorption of alkanes, and separation of CO_2_/N_2_ gas mixtures [1,2,3,4,5,6,7,8]. However, Silicalite-1 zeolites are usually produced under expensive organic template systems [9,10,11], resulting in high production costs that make it difficult to produce on a large scale. In addition, the generation of hazardous gases and organic wastewater during calcination and post-treatment can cause serious environmental pollution. Therefore, it is crucial to develop new technology to reduce the use of templates, either from the point of view of industrialization and application or green development.

To date, template-free and low-template methods have been proposed. For the template-free method, the absence of an organic template in the system increases the activation energy for nucleation and crystal growth, slows down the reaction rate, and thus, the crystallization time was significantly prolonged [12,13]. Furthermore, this method also suffers from the shortcomings of harsh conditions and a narrow phase region [14,15], which is thereby not conducive to industrialized production. On the contrary, the low-template method is an economical way to produce zeolites in the synthetic industry for high efficiency and high crystallization. A number of reports have proposed compensating the shortcomings of the low-template agent synthesis of zeolites by introducing substances such as fluoride [16,17], crystalline seeds [18,19,20], and ethanol [21,22]. Wang et al. [23] introduced Silicalite-1 crystals into the system with a TPA^+^/Si ratio of 0.1, and microporous Silicalite-1 zeolites were obtained after 20 h. The addition of crystal seeds improved the crystallinity of the zeolites and changed the morphology of zeolite. Wu et al. [24] exhibited that Silicalite-1 zeolites could be successfully synthesized in the presence of crystal seeds and ethanol template-free under 48 h. The addition of ethanol could only fill the zeolite pores and run out from the zeolite pores at room temperature, thus reducing environmental pollution. By contrast, fluoride, as a mineral, not merely plays a guiding role but plays a booster role for the growth of zeolite skeleton, among which NH_4_F can control the pH in the system very well. In the NH_4_F system, MFI-type (containing Si and Al) zeolites were synthesized with TPA^+^/Si ratios of 0.048 and 0.1 through 72 h, respectively [25,26]. The introduction of NH_4_F accelerated the formation of the zeolite skeleton and inhibited the entry of Al, which increased the Si/Al ratio in the zeolite structure. However, longer crystallization time is required after the introduction of additives, which causes high energy consumption. Few studies have been conducted on the synthesis of Silicalite-1 zeolites with a TPA^+^/Si ratio less than 0.01.

Owing to the unique and excellent properties of Silicalite-1 zeolites, it is of great significance to explore a green method for the synthesis of zeolites with low-template (TPA^+^/Si = 0.007) and a short crystallization time. In addition, the crystallization mechanism of NH_4_F and S-1 seeds was investigated in detail. It was experimentally found that NH_4_F and S-1 seeds synergistically control the morphology and size of the zeolites, which not only provides new insights into the controllable synthesis of zeolites at the microscopic level but also offers theoretical justification for the large-scale production of zeolites by the low-template method.

## 2. Materials and Methods

### 2.1. Materials

The synthesis of both the seed and product zeolites was performed with Silica gel (SiO_2_, 40%, Qingdao Mack Silica Gel Desiccant Co., Ltd., Qingdao, China), tetrapropylammonium bromide (TPABr, 98%, Maclean’s Biochemical Technology Co., Ltd., Shanghai, China), sodium hydroxide (NaOH, AR, Chengdu Kelong Chemical Co., Ltd., Chengdu, China), and Ammonium fluoride (NH_4_F, AR, 98%, Maclean’s Biochemical Technology Co., Ltd., Shanghai, China). The deionized water (H_2_O) was prepared in the laboratory. All reagents can be used without further purification.

### 2.2. Methods

The seeds were prepared with a molar ratio of 1 SiO_2_: 0.057 TPABr: 0.24 NaOH: 13.66 H_2_O. The synthesis process is as follows: silica gel was dissolved in the deionized water. Then, TPABr was added to the above solution and stirred for 1 h. After that, sodium hydroxide solution was added dropwise into the mixture above and continuously stirred at room temperature for 3 h. The mixture was then transferred to the Autoclave equipped with Teflon liners and subjected to hydrothermal treatment at 180 °C for 8 h. Then, the solid was centrifuged and washed with distilled water. The obtained product was dried at 120 °C for 3 h, and the colloid was used as a seed (named S-1 seeds).

Silicalite-1 zeolites were prepared from a hydrogel of the composition 1 SiO_2_: 0.007 TPABr: x NH_4_F: 0.24 NaOH: 13.66 H_2_O: y S-1 seeds (y: percent of total system mass). The silica sol was put into water. Then, TPABr was added and stirred for 30 min. NH_4_F or S-1 seeds were added to the above solution and further stirred for 30 min. Then, sodium hydroxide was dissolved in distilled water, and the sodium hydroxide solution was dropped into the above mixture. The reaction mixture is stirred continuously at room temperature for 3 h to reach a homogenous state. The gels were heated statically at 180 °C for 8 h. Then, the solids were centrifuged, thoroughly washed, and dried at 120 °C in air. The obtained samples are recorded as S1-x-y, where x refers to the molar ratio of F/Si (x = 0–0.4), and y represents the mass fraction of S-1 seeds in the total system (y = 1–20 wt%).

### 2.3. Characterization

X-ray diffraction is a research means and method to analyze the internal structure of samples and is often used as a basic method to characterize zeolites. Currently, the equipment is commonly used for the diffraction position of the characteristic peaks, intensity, and lattice parameters. In this paper, the crystal phase of Silicalite-1 zeolites was analyzed and characterized using X-ray diffraction (XRD, Shimadzu XRD-6000, Shimadzu, Tokyo, Japan, Cu-Kα radiation). This was used as a means of determining the relative crystallinity of products and thus obtaining the appropriate amount of raw material to be added. The scanning 2θ range was from 5 to 60°, and the scanning rate was 8° min^−1^. The crystallinity of the samples was determined from the sum of the intensities (height) of the major peaks located at 2θ = 7.9, 8.9, 23.0, 23.9, and 24.4°. The S-1 seed was used as the basic standard, defined as 100%, and the ratio of the crystallinity of the other samples compared to the S-1 seed is the relative crystallinity (abbreviated as *RC*). Its calculation formula is as follows:*RC* = *R*_1_/*R*_2_ × 100%

In the formula, *R*_1_ represents the sum of the intensities (height) of the major peaks (located at 2θ = 7.9, 8.9, 23.0, 23.9, and 24.4°) value of the synthesized sample; *R*_2_ represents the sum of the intensities (height) of the major peaks (located at 2θ = 7.9, 8.9, 23.0, 23.9, and 24.4°) value of the S-1 seed; *RC* is relative crystallinity. Additionally, the relative intensities of SiO_2_ peaks (abbreviated as SRV) were determined from the sum of the intensities (height) of the major peaks located at 2θ = 5.7, 25.78, 26.94, and 28.24°. The S1 sample was used as the basic standard.

SEM is a common means of characterization to understand the microscopic morphology of samples, the operation of which is to take a few milligrams of sample powder sprinkled on the surface of a short rod made of aluminum and dispersed, and then use focused, high-energy electron beams to scan the surface of the test samples to obtain the physical information, and after further reception and amplification, to show the sample molecular sieves of the microscopic surface morphology. Now, the instrument is commonly used to determine the particle shape and surface characteristics of the sample. Thermo Fisher Scientific (Waltham, MA, USA), model Apreo 2 SEM instrument from the United States was used to observe the morphology and surface characteristics of the samples, with a voltage of 20 kV.

Nitrogen adsorption–desorption analysis is a frequently used means of characterizing porous materials. Nowadays, it is usually used to determine the specific surface area of the zeolite, the pore size of the sample, the pore volume of the sample, and a series of data information so as to analyze the performance of the zeolite. Nitrogen adsorption–desorption experiments at 77 K were conducted in an Autosorb instrument (Mack, Greensboro, NC, USA, model ASAP 2420). The total surface (S_BET_) was calculated according to the (Brunauer–Emmet–Teller) BET isothermal equation, and the total volume (V_total_) was determined from the nitrogen adsorbed volume at P/P_0_ = 0.990. Both the micropore surface area (S_micro_) and the micropore volume (V_micro_) were calculated by the application of the t-plot method. The external surface area (Sext) and mesopore volume (V_meso_) were obtained by the calculated total data minus the corresponding micropore data.

XRF is a complementary technique that presents information about the composition of the elements inside the sample under study. In this paper, the elemental composition of the samples was quantitatively analyzed at 50 KV using a Zetium type XRF instrument from Alemlo, NLD (helium for system flushing and a Pa target for the X-ray tube) manufactured by Panaco. 

## 3. Results and Discussion

### 3.1. Synthesis of Silicalite-1 in Only NH_4_F or S-1 Seeds System

Fundamental work and preliminary explorations have been undertaken to investigate the effect of the addition of NH_4_F or S-1 seeds on the synthesis of Silicalite-1 zeolites. Figure 1a,e show XRD patterns and SEM images of S-1 seeds. Evidently, a spherical product with the size of 7.41 μm was well crystallized (RC = 100%). In the system without the addition of NH_4_F or S-1 seeds (molar ratio of 1 SiO_2_: 0.007 TPABr: 0.24 NaOH: 13.66 H_2_O), the barrel-shaped Silicalite-1 zeolite was obtained (RC = 28.22%), while the characteristic peaks of the SiO_2_ at 5.7, 25.78, 26.94 and 28.24 °are strong (SRV = 100%) (Figure 1b,f). After the introduction of only NH_4_F (F/Si = 0.1), a tetragonal Silicalite-1 sample with RC of 37.47% and SRV of 70.38% was produced (Figure 1c,g). The zeolites obtained were microporous (Figure 1i), and the surface area and microporous volume were 66.65 m^2^/g and 0.03 cm^3^/g, respectively (Table 1). After the S-1 seeds (10 wt%) were added to the system, a higher RC (76.32%) product with spherical morphology was obtained (Figure 1d,h). However, there are still distinct SiO_2_ characteristic peaks in the XRD patterns (SRV = 58.98%). The synthesized zeolite presents a mesoporous structure with a BET surface area of 164.43 m^2^/g and an external surface area of 55.44 m^2^/g (Figure 1j and Table 1). The results indicated that the single addition of NH_4_F or S-1 seeds could promote the formation of a zeolite skeleton, but none of them can completely inhibit the SiO_2_ heteropoly. Moreover, the addition of S-1 seeds is beneficial for generating mesoporous structures, which can reduce transfer resistance and improve the coke capacity.

To further understand the role of NH_4_F and S-1 seeds in the synthesis of Silicalite-1 zeolites, a systematic study was carried out. The influence of NH_4_F content solely on Silicalite-1 zeolites was investigated by keeping a mixture of 1 SiO_2_: 0.007 TPABr: x NH_4_F: 0.24 NaOH: 13.66 H_2_O, where x varied between 0 and 0.2. All samples displayed a characteristic MFI-type peak (Appendix A). Although the RC of the products first increased and then decreased with the increase of NH_4_F content, the amounts of SiO_2_ significantly decreased in parallel (Figure 2a). This might be ascribed to the fact that the increase of F^−^ concentration promotes the dissolution rate of the silica source. Notably, it obtained a product with RC of 37.47% at x equal to 0.1, which is significantly better than the amorphous product obtained by Sanhood et al. (F/Si = 0.243) [27]. Therefore, the amount of NH_4_F is one of the important factors in the synthesis of zeolites. On the opposite end, the SRV decreases dramatically from 100% (F/Si = 0) to 0 (F/Si = 0.15 and 0.2), indicating that the addition of F^−^ plays a critical role in inhibiting SiO_2_ while exhibiting little effect in improving the crystallinity of zeolites. This result is consistent with that reported by Wang Chang et al. [23]. It is noteworthy that the barrel-shaped crystals obtained in the NH_4_F-free system (Figure 1f) turned into tetragonal crystals (Appendix A) as the NH_4_F content increased. This is due to the fact that F^−^ is first adsorbed on the (010) face and then combines with SiO_2_ to form [SiO_4/2_F]^−^ units, which controls the growth of TPA^+^ along a specific direction, leading to a change in the morphology of the obtained crystals [16].

The effect of S-1 seed content on Silicalite-1 zeolites was explored by assigning S-1 seeds (1 SiO_2_: 0.007 TPABr: 0.24 NaOH: 13.66 H_2_O: y S-1 seeds, y = 0–0.15). As shown in Figure 2c–e, in the single seeds system, the introduction of S-1 seeds could induce the rapid formation of nuclei and reduce the size of the Silicalite-1 crystal with the increase of S-1 seeds content. The same phenomenon was reported by Dai et al. [18]. Moreover, the increase in S-1 seeds content resulted in a sharp rise in the RC of products from 28.22% to 76.32%, and then slightly decreased to 64.30% (Figure 2a,b,f). However, the SRV decreased from 100% to 54.65%, which is quite different from the NH_4_F system. The S-1 seeds can mainly improve the RC of zeolites. Moreover, product yields ranged from 99.32% to 94.15%, which is higher than that of S1 (93.00%). Based on the foregoing, the addition of S-1 seeds can effectively rise up the yield of the zeolites. Surprisingly, the addition of the S-1 seed resulted in the transformation of the Silicalite-1 crystal from a tonneau shape to a spherical shape, which is similar to that of the S-1 seed (Figure 1f and Figure 2c–e), indicating that the morphology of the seed plays a key role in the synthesis of Silicalite-1 crystals.

### 3.2. Impact of the Synergistic Action of NH_4_F and S-1 Seeds

The above results show that the NH_4_F is a key factor in inhibiting the formation of SiO_2_, and the S-1 seeds are critical in improving the RC of the product. Nevertheless, neither the NH_4_F system nor the S-1 seeds system can enhance crystallinity and inhibit SiO_2_ at the same time. Hence, the synergistic synthesis of zeolites with NH_4_F and S-1 seeds was expected to obtain zeolites with pure phase and high crystallinity.

System with a mixture of 1 SiO_2_: 0.007 TPABr: 0.1 NH_4_F: 0.24 NaOH: 13.66 H_2_O: y S-1 seeds was examined, where y ranged from 1 wt% to 20 wt%. In this system, the product yields from 99.13% to 110.10%. The RC of the product increases continuously with the increase of the seed content, whereas the SRV ranges from 56.39% (y = 1 wt%) to 0 (y = 10 wt%) and finally to 10.41% when y = 20 wt% (Figure 3e and Appendix A), which indicates a critical factor of the seed amount for obtaining pure phase Silicalite-1 zeolites. Remarkably, a novel finding was uncovered via SEM images that the Silicalite-1 crystal shape is tetragonal, and no spherical crystals appear when y = 1 wt% (i.e., NH_4_F content is approximate or even higher than the S-1 seed content) (Figure 3a and Appendix A) and the tetragonal crystals disappeared, and spherical crystals dominated when the seed content was increased from 5 wt% to 20 wt% (i.e., the NH_4_F content was lower than the S-1 seed content) (Figure 3b–d and Appendix A). The discovery of this new phenomenon has great potential for adjusting the morphology of zeolites.

A system with a mixture of 1 SiO_2_: 0.007 TPABr: x NH_4_F: 0.24 NaOH: 13.66 H_2_O: 10 wt% S-1 seed was studied, where x was between 0.03 and 0.4. As the amount of NH_4_F increased, the yield of the product consistently rose from 77.74% to 95.63%, with a 94.91% yield for S1-0.1–10 wt%. Moreover, the elemental analysis of this product was carried out by XRF, and it was found that there was no elemental Al detected within the range of detection, only Si. While the MFI-type peak intensity of the product first increased and then decreased (Figure 3f and Appendix A). As speculated above, each product was devoid of SiO_2_ peaks (Figure 3f and Appendix A). Strikingly, all Silicalite-1 crystals are spherical (Appendix A), which further confirms that the Silicalite-1 crystal morphology is determined by the seed morphology when the amount of NH_4_F is less than that of the S-1 seed.

### 3.3. The Insight Role of NH_4_F and S-1 Seeds

A mixed solution of 1 SiO_2_: 0.007 TPABr: 0.1 NH_4_F: 0.24 NaOH: 13.66 H_2_O was used to investigate the synthesis mechanism of NH_4_F on Silicalite-1 zeolites. A large number of amorphous appeared at 2 h (Figure 4a,b and Appendix A). When the crystallization time was 4 h, a few tetragonal crystals emerged (Figure 4a,b and Appendix A). The tetragonal crystals increased further at 6 h (Figure 4a,b and Appendix A), and at 8 h, the number of tetragonal crystals continued to increase, the amorphous state disappeared, and SiO_2_ appeared (Figure 4a,b). According to these results, Figure 1a shows the evolution of Silicalite-1 crystals in this system.

To understand how the seeds promote the growth of Silicalite-1 zeolites, a mixture of 1 SiO_2_: 0.007 TPABr: 0.24 NaOH: 13.66 H_2_O: 10 wt% S-1 seeds was investigated. As shown, at the early stage, the S-1 seeds were spherical and concentrated (Figure 1e), and partial S-1 seeds dissolved into a gel at 2 h (Appendix A and Figure 4c,d). When the crystallization time reached 4 h, the remaining S-1 seeds continued to dissolve (Appendix A and Figure 4c,d), and some new spherical crystals with rough surfaces appeared at 6 h (Appendix A and Figure 4d). The new spherical crystals continued to increase, and the surfaces became smooth (Figure 1h and Figure 4d). The SiO_2_ appeared at 8 h. The phenomenon contributes to the principle of dissolution followed by crystallization. However, when the S-1 seeds were dissolved into the amorphous solution, their fundamental units that constitute the MFI-type structure were retained, which in turn facilitated the growth of Silicalite-1 crystals [28,29]. Figure 1b shows the reaction of Silicalite-1 crystals in the single S-1 seed system.

The synthesis mechanism of Silicalite-1 zeolites was executed for a mixture of 1 SiO_2_: 0.007 TPABr: 0.1 NH_4_F: 0.24 NaOH: 13.66 H_2_O: 10 wt% S-1 seeds. Compared to the single S-1 seeds system, the NH_4_F displays unique phenomena that new spherical crystals of Silicalite-1 zeolites already had appeared at 2 h (Figure 4f), attributed to the fact that F^−^ accelerated the dissolution of the S-1 seeds, which led to the rapid dissolution of the S-1 seeds into the gel. In addition, F^−^ balances the charge of the cation in the system, and F^−^ occupies well-defined positions in the 4^1^·5^2^·6^2^ cages where they are covalently bonded to framework Si atoms [30]. The new spherical crystals increased further at 4 h. When the time reached 6 h or even 8 h, the system was saturated with S-1 seeds, the RC did not change significantly, and there was no SiO_2_ in the system (Figure 4e). Further prolonging the time to 12 h, 24 h, and 48 h, the appearance of SiO_2_ can be seen. This is attributed to the fact that the system undergoes transcrystallization and tends to synthesize a more stable crystalline phase at a certain degree in the crystallization time (Appendix A). Therefore, in combination with Section 3.2. and the above results, Figure 2 shows the evolution of Silicalite-1 crystals in this system.

It can be determined that F^−^ can promote the dissolution of silicon sources. While the S-1 seeds it is based on the mechanism of first dissolution and then crystallization. In addition, when the S-1 seeds dissolve into the amorphous state, the basic structural units for constructing the MFI-type are retained, providing a nucleation site for the Silicalite-1 crystal. Meanwhile, the synergistic effect of NH_4_F and S-1 seeds plays a vital role in the synthesis of pure-phase zeolites with high crystallinity.

## 4. Conclusions

This work developed a green and efficient synthesis method that utilizes the synergistic effect of NH4F and S-1 crystal seeds. The process accelerated the formation and growth of nuclei, shortened the crystallization time, reduced energy consumption, and synthesized pure-phase, high-crystallinity zeolites with yields as high as 94.91%. It was found that F^−^ accelerates the dissolution of the silica source and forms [SiO_4/2_F]^−^ units with SiO_2_, which allows the growth of TPA^+^ along a specific direction. In addition, S-1 seeds play a large active role in the skeleton growth of zeolites, compensating for the lack of organic templates in the system. Noticeably, both NH_4_F and S-1 seeds have a significant effect on the crystal morphology of silicalite-1. When the content of NH_4_F was close to or even higher than that of S-1 seed, the crystal morphology of Silicalite-1 zeolites was determined by NH_4_F. When the NH_4_F content is less than the S-1 seeds content, the Silicalite-1 crystal morphology is determined by the S-1 seeds morphology. This work not only provides a new pathway for future industrial-scale production of zeolites using the low-template method but also gives a new insight into the precise regulation of the morphology and size of zeolites, which can be applied to the methanol-to-propylene process.

## Data Availability

Data are contained within the article and Appendix A.

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
