# Peer review of "The Synergistic Impact of Crystal Seed and Fluoride Ion in the Synthesis of Silicalite-1 Zeolite in Low-Template Systems"

_materials, 2024, doi:10.3390/ma17010266_

Round 1

Reviewer 1 Report

Comments and Suggestions for Authors

The authors describe the TPA+ templated synthesis of silicalite-1 zeolites by investigating the effects of the presence of silicate-1 seeds and/or NH4F. They conclude that they have identified a green and efficient route for preparing Silicalite-1 zeolite by a route with a low template concentration (TPA+/Si = 0.007). Due to the presence of NH4F (F/Si = 0.1) and seeds (10 wt%), the product formed in a much shorter time with respect to its formation in their absence and it is also claimed that the NH4F is beneficial for inhibiting the formation of SiO2. XRD, SEM and BET are the characterisation techniques used  

Zeolite synthesis is a well-established procedure and the gains claimed by the authors are somewhat modest. However, the results could be of interest if a revised version of the paper is provided.

The main problem is that it is difficult to follow the arguments being made by the authors, as the tend to draw direct conclusions from the figures, without clearly illustrating how these conclusions are reached.  For example, I find it very difficult to see any significant differences between the SEM images in Figures 2c-e

Similarly, in relation to Figure 4, the authors state: “A large number of amorphous  appeared at 2 h (Fig. 4a and Fig. 4b). When the crystallization time was 4 h, a few of tetragonal crystals emerged (Fig. 4a and Fig. 4b). The tetragonal crystals increased further at 6 h (Fig. 4a and Fig. 4b), and at 8 h, the number of tetragonal crystals continued to increase, the amorphous state disappeared, and SiO2 appeared”

From the images it could be argued that there are in fact more tetragonal crystals after 2 h than after 4 h. At least there seem to be more with sharp edges.

Are what is described as amorphous really amorphous or are there just smaller crystals in the image?

Is should also be remembered that the SEM images only show the area reported and one image is not necessarily indicative of the whole product. 

One action that the authors should take is to identify on the SEM images examples of what they describe as tetragonal crystals and spherical crystals (which are very likely to in fact be polymprphous agglomerates) 

There are a number of phrases in the manuscript which are a little ambiguous. For example in the abstract, the penultimate sentence (lines 19-21) is difficult to understand. This is not a general problem, but occurs occasionally and thr manuscript should be carefully checked.  

The concept of relative crystallinity (RC) should also be clearly explained and indications on how it is calculated should be provide

Comments on the Quality of English Language

There are a number of phrases in the manuscript which are a little ambiguous. For example in the abstract, the penultimate sentence (lines 19-21) is difficult to understand. This is not a general problem, but occurs occasionally and thr manuscript should be carefully checked.

Author Response

Thank you for your suggestion. Based on your suggestion, we have compiled it into a PDF, and the relevant information has been uploaded to the PDF.

Reviewer 2 Report

Comments and Suggestions for Authors

The manuscript Green synthesis of Silicalite-1 zeolites from the synergism strategy between zeolite seeds and fluoride ion presents a new green and efficient route for preparing Silicalite-1 zeolite, a material widely used in industry for selective catalysis, adsorption of alkanes, and separation of gases. The paper is generally well written and the results are of interest.

Below are several recommendations aiming to improve the manuscript:

1.      Abstract: the last sentence should be improved to drawn the main conclusion of this work.

2.      An estimation of the producing costs of  Silicalite-1 zeolites comparing with the existing synthesis methods should be on interest

Author Response

(The authors gave the same response as above.)

Reviewer 3 Report

Comments and Suggestions for Authors

The authors present a paper entitled: "Green synthesis of Silicalite-1 zeolites from the synergism strategy between zeolite seeds and fluoride ion ". After review, the following observations are highlighted:

1.    Title: the authors title their work as follows: "Green synthesis of Silicalite-1 zeolites from the synergism strategy". Authors are required to review and revise the title. The way it is written is very ambiguous and scattered. The title should simultaneously contain the essence of the work (objectives, scope, impact...). Please modify.

2.    Abstract: The way the Abstract is written is not correct. It is recommended that the authors follow a logical order to guide the reader in the meaning of this research. The specific objective of this work should be stated, which methods were used in its development, which results are noteworthy and in which field of science or industry they could be applied. Please rewrite.

3.    Introduction: Zeolite synthesis is an expensive process and in that detail also lies its limitation of use on an industrial scale; however, natural zeolite deposits are very extensive and abundant all over the planet, while their extraction and processing are very cheap. It is recommended that the authors dedicate a paragraph mentioning the advantage of working with synthetic zeolites instead of natural zeolites. Please add.

4.    Introduction: Authors are encouraged to add a paragraph at the end of the Introduction highlighting the importance and novelty of this work. Add.

5.    Section 2. It is recommended to change the word "Experimental" to "Materials and Methods". Change.

6.    Subsection 2.2: It is recommended to rename this subsection as "2.2. Methods"; then all procedures are described below. Change.

7.    Section 2.3. Characterisation: Lines 96 and 104. The description of the technical characteristics and analytical scope of the mentioned equipment are very poor or absent. Please provide more details.

8.    Section 2.3. Characterisation: It is not clear what is intended when characterising the synthesised Silicalite-1 zeolites by XRD. Please explain in more detail.

9.    Lines 118 et seq: The way figures are referred to is incorrect. Authors are advised to use the following style: Figure 1a; Figure 1e; Figure S1; and so on. Please do not write in bold or use abbreviations. The entire manuscript should be proofread and corrected.

10. Line 131. Please do not write the name of the tables in bold. Fix.

11. Section 3. Results and discussion. The authors have presented a detailed set of results obtained, but in reality only these results are described. A proper discussion has not been made since the authors do not contrast the results obtained with those of other authors; in fact, only a single bibliographical reference [27] has been used in Line 143. It is suggested that the authors rewrite this section by adding new and abundant references.

12. The authors have developed a new zeolite (Silicalite-1). The authors are asked to answer the following questions:

a.    What is the percentage of SiO2 in this new mineral?

b.    What is its pore size?

c.    What is its ion exchange capacity?

d.    What is its adsorption capacity?

e.    What is its Si/Al ratio?

f.     What is its loss on ignition (LOI)?

Section 4. Conclusions: the authors should highlight in this section the percentage of Silicalite-1 obtained. On the other hand, it is emphasised that the authors have to reflect the fields of application of these results.

Comments on the Quality of English Language

Moderate editing of English language required.

Author Response

(The authors gave the same response as above.)

Reviewer 4 Report

Comments and Suggestions for Authors

The present paper, entitled "Green synthesis of Silicalite-1 zeolites from the synergism strategy between zeolite seeds and fluoride ion" reports an interesting work on the very important topic of green synthesis of zeolites. The paper summarizes valuable information in its figures and tables, and contains a lot of work. The manuscript needs several revisions before it can be published. Therefore, please improve or clarify the following points:

  1. It is necessary to have an easier code of notation for your sample or to introduce a table where to refer to all codes in the beginning.
  2. It is important to keep the same color code in SEM images for the sample at the same number of hours in the measurements.
  3. Please introduce in Table 1 the values for the other samples used in your research.
  4. Use subscripts in writing the chemical formulas (see references).
  5. The homogeneity of the reference section needs to be improved. The journals should be abbreviated. See : https://www.mdpi.com/journal/materials/instructions

 Based on these, I advise the authors to rectify the above-mentioned errors, and I hope to re-evaluate the revised manuscript.

Author Response

(The authors gave the same response as above.)

Round 2

Reviewer 3 Report

Comments and Suggestions for Authors

The authors have adequately answered all the questions asked and have improved the quality of the manuscript following the suggestions and recommendations made by the reviewer.

Reviewer 4 Report

Comments and Suggestions for Authors

The paper was significantly improved and it is ready to be published in this form.